# DeepEnFM: Deep neural networks with Encoder enhanced Factorization Machine

## Abstract

Click Through Rate (CTR) prediction is a critical task in industrial applications, especially for online social and commerce applications. It is challenging to find a proper way to automatically discover the effective cross features in CTR tasks. We propose a novel framework for CTR task, called Deep neural networks with Encoder enhanced Factorization Machine (DeepEnFM). Instead of learning the cross features directly, DeepEnFM adopts the Transformer encoder as a backbone to align the feature embeddings by the clues of other fields. The embeddings generated from the encoder are beneficial for further feature interactions. Particularly, DeepEnFM utilizes a bilinear approach to generate different similarity functions with respect to different field pairs. Furthermore, the max-pooling method makes DeepEnFM feasible to capture both the supplementary and suppressing information among different attention heads. Our model is validated on the Criteo and Avazu datasets, and achieves the state-of-art performance.

## 1 Introduction

This paper studies the problem of predicting the Click Through Rate (CTR), which is an essential task in industrial applications, such as online advertising, and e-commerce. To be exact, the advertisements of cost-per-click (CPC) advertising system are normally ranked by the eCPM (effective cost per mille), which is computed as the prodcut of bid price and CTR (click-through rate). To predict CTR precisely, feature representation is an important step in extracting the good, interpretable patterns from training data. For example, the co-occurrence of "Valentine's Day", "chocolate" and "male" can be viewed as one meaningful indicator/feature for the recommendation. Such handcrafted feature type is predominant in CTR prediction (Lee et al., 2012), until the renaissance of Deep Neural Networks (DNNs).

Recently, a more effective manner, *i.e.*, representation learning has been investigated in CTR prediction with some works (Guo et al., 2017; Qu et al., 2016; Wang et al., 2017; Lian et al., 2018; Song et al., 2018), which implicitly or explicitly learn the embeddings of high-order feature extractions among neurons or input elements by the expressive power of DNNs or FM. Despite their noticeable performance improvement, DNNs and explicit high order feature-based methods (Wang et al., 2017; Zhang et al., 2016; Guo et al., 2017; Lian et al., 2018) seek better feature interactions merely based on the naive feature embeddings.

Few efforts have been made in addressing the task of holistically understanding and learning representations of inputs. This leads to many practical problems, such as "polysemy" in the learned feature embeddings existed in previous works. For example, the input feature 'chocolate' is much closer to the 'snack' than 'gift' in normal cases, while we believe 'chocolate' should be better paired with 'gift' if given the occurrence input as "Valentine's Day". This is one common polysemy problem in CTR prediction.

Towards fully understanding the inputs, we re-introduce to CTR, the idea of Transformer encoder (Vaswani et al., 2017), which is oriented in Natural Language Processing (NLP). Such an encoder can efficiently accumulate and extract patterns from contextual word embeddings in NLP, and thus potentially would be very useful in holistically representation learning in CTR. Critically, the Transformer encoder has seldom been applied to CTR prediction with the only one exception arxiv paper AutoInt (Song et al., 2018), which, however, simply implements the multi-head self-attention (MHSA) mechanism of encoders, to directly extract high-order feature interactions. We

argue that the output of MHSA/encoder should be still considered as first-order embedding influenced by the other fields, rather than a high-order interaction feature.

To this end, our main idea is to apply the encoder to learn a context-aware feature embedding, which contains the clues from the content of other features. Thus the "polysemy" problem can be solved naturally, and the second-order interaction of such features can represent more meaning. Contrast to AutoInt(Song et al., 2018), which feeds the output of encoder directly to the prediction layer or a DNN, our work not only improves the encoder to be more suitable for CTR task, but also feeds the encoder output to FM, since both our encoder and FM are based on vector-wise learning mechanism. And we adopt DNN to learn the bit-wise high-order feature interactions in a parallel way, which avoids interweaving the vector-wise and bit-wise interactions in a stacked way.

Formally, we propose a novel framework – Deep neural networks with Encoder enhanced Factorization Machine (DeepEnFM). DeepEnFM focuses on generating better contextual aligned vectors for FM and uses DNN as a bit-wise information supplement. The architecture adopting both Deep and FM part is inspired by DeepFM (Guo et al., 2017). The encoder is endowed with bilinear attention and max-pooling power. First, we observed that unlike the random order of words in a sentence, the features in a transaction are in a fixed order of fields. For example, the fields of features are arranged in an order of {Gender, Age, Price ...}. When the features are embedded in dense vectors, the first and second vectors in a transaction always represent the field "Gender" and "Age". To make use of this advantage, we add a bilinear mechanism to the Transformer encoder. We use bilinear functions to replace the simple dot product in attention. In this way, feature similarity of different field pairs is modeled with different functions. The embedding size in CTR tasks is usually around 10, which allows the application of bilinear functions without unbearable computing complexity. Second, the original multi-head outputs are merged by concatenation, which considers the outputs are complementary to each other. We argue that there are also suppressing information between different heads. We apply a max-pooling merge mechanism to extract both complementary and suppressing information from the multi-head outputs. Experimental results on Criteo and Avazu datasets have demonstrated the efficacy of our proposed model.

To summarize, our contributions are three-fold. (1) We propose a novel model DeepEnFM, which combines both DNN and encoder enhanced FM. (2) The state-of-the-art results on Criteo and Avazu datasets have shown the efficacy of our proposed model. (3) Extensive ablation study has shown the contribution of each component, and revealed the relations between DNN, encoder, and FM by developing different architectures.

## 2 RELATED WORK

CTR prediction is a critical task in Recommendation System (Adomavicius and Tuzhilin, 2005; Batmaz et al., 2019), which aims to predict the probability of a user click behavior based on the given item. The key challenge of the CTR is to automatically learn combinatory/cross features (e.g., a 2-way feature: "Valentine's Day" and "chocolate", a 3-way feature: "Valentine's Day", "chocolate" and "male") from the inputs.

Traditional learning methods, such as Logistic Regression (LR)(Lee et al., 2012), learn weights of processed features in supervised way for prediction. To gain a better performance, the manually designed combinatory features are fed into LR model, which is laborious and unable to cover the numerous combination of features.

To learn the combinatory features automatically in a explicit way, Rendle proposes Factorization Machine (Rendle, 2010) model, which treats the weight of a feature pair as the dot of two latent vectors. Moreover, HOFM (Blondel et al., 2016) and CIN (Lian et al., 2018) are proposed to capture the interaction of arbitrary order in a explicit manner. However, the primary/first-order field embeddings are utilized without considering the contextual information, which leads to the 'polysemy' problem in field embedding unsolved.

Deep Neural Networks (DNN) (Krizhevsky et al., 2012) have made remarkable progresses in CV and NLP tasks by its extraordinary representation learning power. Several DNN based CTR models are developed on the basis of primary field embeddings (Zhang et al., 2016; Cheng et al., 2016; Guo et al., 2017; Lian et al., 2018; Wang et al., 2017) or product layers (Qu et al., 2016) with a plain DNN. One of the major drawbacks of the plain DNN is that it tackles the field embeddings as a large

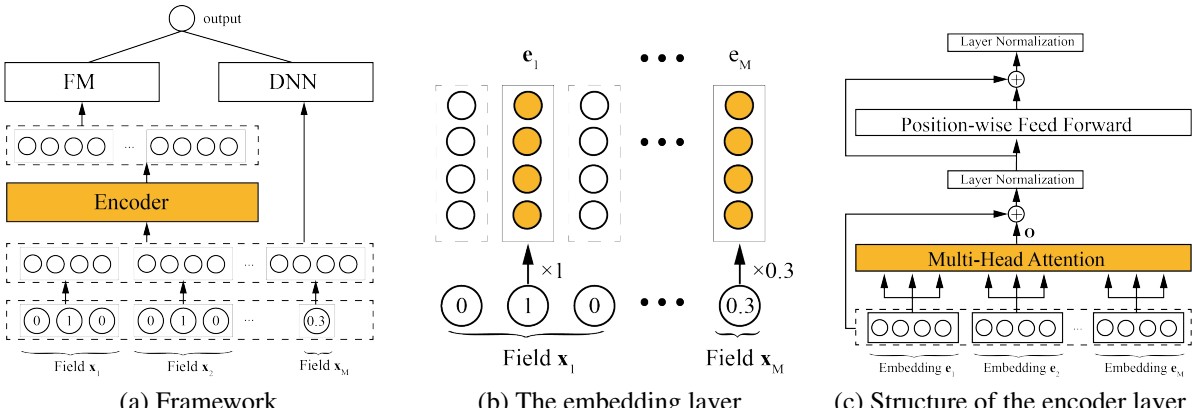

Figure 1: (a) The framework is mainly composed of encoder, DNN and FM. (b) The embedding layer maps both categorical fields and numerical fields into dense vectors. (c) The encoder layer is composed of multi-head self-attention module (MHSA) and position-wise feed forward module with the layer normalization and residual connection.

flat layer, which means there are only interactions between neurons. Thus, DNN can be utilized as a bit-wise interaction feature learner to assist the vector-wise learner.

Recently, Attention mechanism, which has been widely used in Machine Translation (Sutskever et al., 2014) and Image Caption (Xu et al., 2015) tasks, has been adopted in CTR models (Xiao et al., 2017; Song et al., 2018). Specially, Transformer (Vaswani et al., 2017) encoder, empowered with multi-head self-attention, is demonstrated to be capable of generating context-aware word embeddings in the work BERT (Devlin et al., 2018). Therefore, it seems promising to use Transformer encoder to solve the 'polysemy' problem in CTR. Thus, we propose our model DeepEnFM to employ the encoder for context-aware field embedding. A related work to our model is AutoInt (Song et al., 2018). However, AutoInt only utilizes the multi-head self-attention of encoder for high-order feature extraction rather than for field embedding.

## 3 METHODS

**Task Formulation**. The CTR task is to predict the click behavior from a record of multiple fields. Assume we have a training record as $\{\mathbf{x}, y\}$, where $\mathbf{x}$ represents the fields of instance and $y \in \{0, 1\}$ represents the click behavior. The instance fields $x$ may contain both categorical fields and numerical fields. We denote $y = 1$ when the click behavior is 'click'; otherwise, $y = 0$. Given a unseen test record $x^*$, the goal is then to learn a mapping function $y^* = \Psi(x^*)$ using all available training information and predict the probability of user click behavior $y^*$, where the $y^* \in [0, 1]$ represents the probability of click behavior.

---

**Algorithm 1** Algorithm of DeepEnFM

1: $\mathbf{E}^{(0)} = \text{Embedding}(\mathbf{x})$
2: **for** $l \leftarrow 1, L$ **do**
3:     $\mathbf{E}^{(l)} = \text{Encoder}(\mathbf{E}^{(l-1)})$
4: **end for**
5: $\mathbf{y}_{\text{DNN}} = \text{DNN}(\text{Flatten}(\mathbf{E}^{(0)}))$
6: $\mathbf{y}_{\text{FM}} = \text{FM}(\mathbf{E}^{(L)})$
7: $\hat{y} = \text{Prediction}(\mathbf{y}_{\text{DNN}}, \mathbf{y}_{\text{FM}})$

---

**Overview.** Our framework DeepEnFM is illustrated in Fig. 1(a), which is composed of five components: the embedding layer, encoder, FM, DNN and prediction layer. The workflow is depicted in Alg. 1. Firstly, the embedding layer projects the input fields into a low dimensional space. Then the encoder adaptively updates the embedding results with respect to the clues from other fields through the multi-head self-attention module (MHSA). Next, FM calculates a score which integrates the

explicit interactions between encoder outputs. Simultaneously, the original results from the embedding layer are fed into DNN to learn implicit feature interactions at bit-wise level. Finally, we apply the prediction layer to map the intermediate outputs of the FM and DNN into the final score with sigmoid function.

### 3.1 THE COMPONENTS OF DEEPENFM

**Embedding.** The embedding layer converts the instance input $\mathbf{x}$ from a high dimensional space to a low dimensional, dense vector. As shown in Fig. 1(b), we denote the record input fields as $\mathbf{x} = [\mathbf{x}_1, \mathbf{x}_2, \ldots, \mathbf{x}_M]$, where $\mathbf{x}_i$ can be either a one-hot vector (categorical field) or a scalar (numerical field). We assume that the field embedding size is $d$, and the output of embedding layer can be arranged as a matrix $\mathbf{E} = [\mathbf{e}_1; \mathbf{e}_2; \ldots; \mathbf{e}_M]$. For a categorical field, the field value is used to look up the corresponding embedding vector as the token index in word embedding. For a numerical field, the field index is used to look up the related embedding vector, and the field value impacts the embedding as a factor.

**Encoder**. The encoder is a stack of $L$ encoder layers with residual connection, which is designed to align the field embedding results accumulatively according to the other field contents. The output of L-th layer can be denoted as $E^{(L)}$. As shown in Fig. 1(c), the encoder layer contains two modules: a multi-head self-attention module and a position-wise feed-forward module. At the end of each module, there is a residual connection followed by the layer normalization. We will further explain these two modules in Sec. 3.2.

**FM**. FM is leveraged to explicitly capture the first order and second order feature interactions as a routine in practice. In our model, FM calculates the score based on the encoder outputs $E^{(L)}$, which are the context-aware field embeddings.

$$y_{\text{FM}} \quad = \quad \sum_{i=1}^{M} \mathbf{w}_i \mathbf{x}_i + \sum_{i=1}^{M} \sum_{j=i+1}^{M} \mathbf{e}_i^{(L)} \cdot \mathbf{e}_j^{(L)} \tag{1}$$

**DNN.** The DNN aims to capture the implicit high-order interactions between different fields in the bit-wise level. The output of embedding layer is flattened to a large vector. Then the large vector is fed into a multiple-layer feed-forward neural network to extract features.

$$y_{\text{DNN}} \quad = \quad \text{DNN}(\text{Flatten}(\mathbf{E}^{(0)})) \tag{2}$$

We do not use the output of encoder as the input of DNN, because both the encoder and FM learn at vector-wise level, while DNN aims at bit-wise level. Sharing encoder input may jeopardize the performance of encoder.

**Prediction**. The prediction layer is a full connection layer, using a simple logistic regression function based on the outputs of FM and DNN to predict the final result.

$$y \quad = \quad \sigma([\mathbf{y}_{\text{FM}}, \mathbf{y}_{\text{DNN}}]\mathbf{w}_p + b_p) \tag{3}$$

### 3.2 THE DETAILS OF ENCODER

**Multi-head Self-attention Module (MHSA).** The procedure of MHSA (Vaswani et al., 2017) is described in Alg. 2 and Fig. 2(a). Suppose the number of heads is $h$ for each head of MHSA, it has the following steps. (1) The input vector set $\mathbf{E}$ is mapping into Query $\mathbf{Q}$, Key $\mathbf{K}$ and Value $\mathbf{V} \in R^{m \times \frac{d}{h}}$. (2) The similarity matrix $\mathbf{S}$ is calculated by each pair of query $\mathbf{q}$ and key $\mathbf{k}$ vectors. (3) The output is calculated as a weighted sum of value vectors $\mathbf{v}$ according to $\mathbf{S}$. Finally, the outputs of different attention heads are merged to the final output.

As shown in Alg. 2, the similarity and merge functions are the key parts of MHSA. The design principle of similarity and merge function mainly depends on the generalization, efficiency and architecture requirements. We will discuss the different implements between MHSA in Transformer and DeepEnFM encoder. As for MHSA in Transformer encoder, we firstly introduce the similarity function in Transformer encoder as shown in Eq. 4. The similarity function is a scaled dot function, which computes the similarity between different feature pairs equally. It merges the output vectors by a concatenation and a linear transformation with $\mathbf{W}^O \in R^{Hd_v \times d}$.

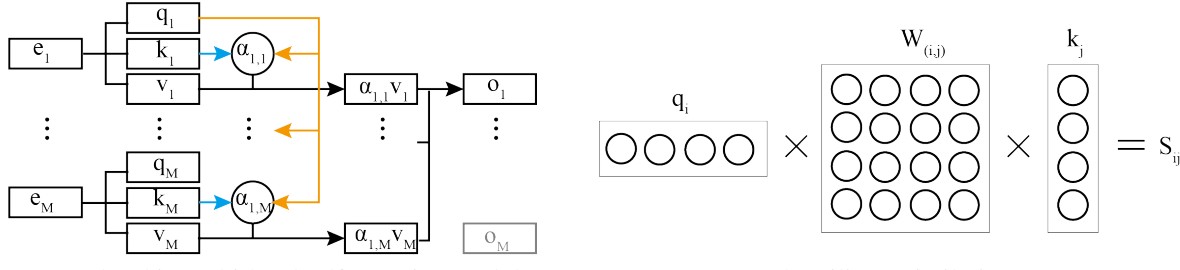

(a) One head in multi-head self-attention module  (b) Bilinear similarity

Figure 2: (a) The procedure for one head in MHSA includes vector mapping, similarity computing and output vector generation. (b) The bilinear function used in MHSA of DeepEnFM encoder.

---

**Algorithm 2** Algorithm of multi-head self-attention

---

1: **for** $h \in \{1, \ldots, H\}$ **do**
2:      $\mathbf{Q}^{(h)}, \mathbf{K}^{(h)}, \mathbf{V}^{(h)} \leftarrow \mathbf{EW}_Q^{(h)}, \mathbf{EW}_K^{(h)}, \mathbf{EW}_V^{(h)}$
3:      **for** $i, j \in \{1, \ldots, M\}$ **do**
4:          $\mathbf{q}_i^{(h)}, \mathbf{k}_j^{(h)}, \mathbf{v}_j^{(h)} \leftarrow \mathbf{Q}_{i,:}^{(h)}, \mathbf{K}_{j,:}^{(h)}, \mathbf{V}_{j,:}^{(h)}$
5:          $\mathbf{S}_{i,j}^{(h)} \leftarrow \text{Similarity}(\mathbf{q}_i^{(h)}, \mathbf{k}_j^{(h)})$
6:          $\alpha_{i,j} = \frac{\exp(\mathbf{S}_{i,j}^{(h)})}{\sum_{j=1}^{M} \exp(\mathbf{S}_{i,j}^{(h)})}$
7:          $\mathbf{O}_{i,:}^{(h)} \leftarrow \sum_{j=1}^{M} \alpha_{i,j} \mathbf{v}_j^{(h)}$
8:      **end for**
9: **end for**
10:   $\mathbf{O} = \text{Merge}(\mathbf{O}^{(1)}, \ldots, \mathbf{O}^{(H)})$

---

$$\text{Similarity}_{\text{sdot}}(\mathbf{q}_i^{(h)}, \mathbf{k}_j^{(h)}) = \frac{(\mathbf{q}_i^{(h)})^{\text{T}} \mathbf{k}_j^{(h)}}{\sqrt{d_k}} \tag{4}$$

$$\text{Merge}_{\text{cat}}(\mathbf{O}^{(1)}, \mathbf{O}^{(2)}, \ldots, \mathbf{O}^{(H)}) = \text{concat}\left(\mathbf{O}^{(1)}, \mathbf{O}^{(2)}, \ldots, \mathbf{O}^{(H)}\right) \mathbf{W}^O \tag{5}$$

**MHSA in DeepEnFM Encoder**. To be clear, we argue that the similarity function between different field pairs could be personalized since the field order is fixed and the field embedding size is very low in CTR. Thus we customize the similarity function for each field pair query $\mathbf{q}_i^{(h)}$ and key $\mathbf{k}_j^{(h)}$ by performing a bilinear function with a specified weight matrix $\mathbf{W}_{(i,j)}^{(h)}$, as shown in Fig. 2(b). Second, we believe that different heads extract features from different viewpoints, so the results contain both complementary and suppressing information to each other. Instead of concatenating all of the head outputs, we keep $\mathbf{V} \in R^{m \times d}$ with the same dimensions of $\mathbf{E}$, and use a max-pooling to get the salient output $\mathbf{O}$. Further, we take the output $\mathbf{O}$ of MHSA as the residual as shown in Fig. 1 (c).

$$\text{Similarity}_{\text{bilinear}}(\mathbf{q}_i^{(h)}, \mathbf{k}_j^{(h)}) = \frac{(\mathbf{q}_i^{(h)})^{\text{T}} \mathbf{W}_{(i,j)}^{(h)} \mathbf{k}_j^{(h)}}{\sqrt{d_k}} \tag{6}$$

$$\text{Merge}_{\text{maxpool}}(\mathbf{O}^{(1)}, \mathbf{O}^{(2)}, \ldots, \mathbf{O}^{(H)}) = \text{Maxpool}\left(\mathbf{O}^{(1)}, \mathbf{O}^{(2)}, \ldots, \mathbf{O}^{(H)}\right) \tag{7}$$

**Position-wise Feed-forward Module**. The position-wise/field-wise feed-forward module (Vaswani et al., 2017) is a one-hidden-layer neural network that firstly maps the field embedding to a higher dimensional embedding space, then back to the original dimensional space.

## 3.3 THE ARCHITECTURE OF DEEPENFM

The encoder, FM and DNN in our DeepEnFM can be combined in parallel or stacked way, we explore different architectures for ablation study. We consider that the encoder can serve as a em-

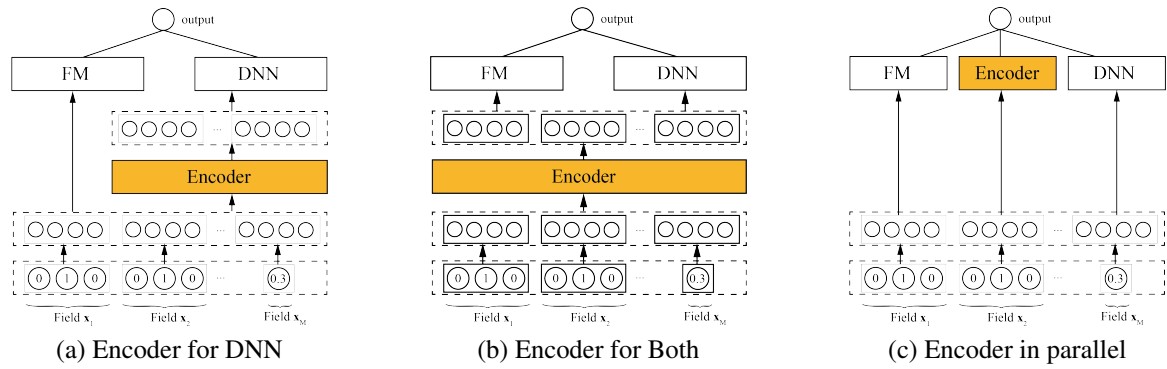

(a) Encoder for DNN     (b) Encoder for Both     (c) Encoder in parallel

Figure 3: Encoder serves in different positions. (a) Encoder serves as the feature extractor for DNN. (b) Encoder serves as the feature extractor for both FM and DNN. (c) Encoder serves as a parallel branch for prediction.

bedding layer for the downstream task, as the BERT in NLP tasks (Devlin et al., 2018). Specially, we stack the encoder and FM together as the left branch as shown in Fig. 1(a), since both of them learn at vector-wise level. Three different architectures are described in Fig. 3, which will be further discussed in ablation study.

## 4   EXPERIMENTS

We conduct experiments on two datasets.

**(1) Criteo dataset.** It contains 45 million click records. Each record includes 13 dense features and 26 categorical ones. For a fair comparison with the other methods, we follow the standard split here: 80%, 10% and 10% of the data as the train, validation and test data respectively. **(2) Avazu dataset**. It contains 40 million click records, 23 fields totally. We also randomly select 80%, 10% and 10% data for train, validation and test respectively. The categorical features with the occurrence more than 10 and 5 times are preserved for Criteo and Avazu respectively, others are set to 'Unknown' as the data processing in AutoInt (Song et al., 2018).

**Evaluation.** The validation data is used to search for hyper-parameter setting. When the hyper-parameters determined, both the the train and validation data are used for training, and the performance is evaluated on the test data. The experiment result is evaluated on AUC (Area Under ROC) and Logloss (Cross Entropy).

**Setting.** We implement our approach by TensorFlow. We choose Adam as optimizer and the learning rate is 0.001. We set the L2 regularization $\lambda$ as 0.0001 and dropout rate as 0.5. The batch size, embedding size, and DNN layer size is set to 2048, 10 and 400 respectively. We use Xavier initialization for dense layers and random normal initialization for bilinear attention weights. The DeepEnFM model is trained on V100 for 1 hours per training epoch.

**Competitors.** We compare our model against several state-of-the-art methods: **(1) LR** (Lee et al., 2012). LR simply learns weights of first-order features. **(2) FM** (Rendle, 2010). FM learns a latent vector for each feature and models feature interactions by dot products of these vectors. This model learns both first and second-order features. **(3) DNN** (Zhang et al., 2016). DNN takes embedded dense vectors as its input and learns high-order features in an implicit way. **(4) IPNN** (Qu et al., 2016). IPNN feeds the dense embeddings to a inner product layer firstly. Then the output is fed to a DNN. **(5) OPNN** (Qu et al., 2016). OPNN replaces IPNN's inner product layer with an outer product layer. This replacement reduce the complexity of the model. **(6)PNN\*** (Qu et al., 2016). PNN\* use the product layer to concatenate the inner and outer product. **(7) DeepFM** (Guo et al., 2017). DeepFM introduces an embedding layer to get dense latent vectors for FM. Then these dense vectors are also fed to a DNN in parallel. **(8) AutoInt** (Song et al., 2018). AutoInt feeds the output of its embedding layer to a interacting layer. Then MHSA based interacting layer can learn high-order features, and a sigmoid output layer or DNN is applied after the interacting layer. **(9) DeepEnFM**. Our model employs a encoder-enhanced FM with a DNN.

Table 1: The results of different models on Criteo and Avazu dataset. DNN depth represents the layers of neural networks. ATT depth represents the layers of MSHA or encoder.

| Model | Criteo | | Avazu | | Depth | |
|---|---|---|---|---|---|---|
| | AUC ($\uparrow$) | Logloss ($\downarrow$) | AUC ($\uparrow$) | Logloss ($\downarrow$) | DNN | ATT |
| LR | 0.7839 | 0.4655 | 0.7688 | 0.3880 | - | - |
| FM | 0.7860 | 0.4672 | 0.7838 | 0.3779 | - | - |
| DNN | 0.7942 | 0.4560 | 0.7731 | 0.3841 | 2 | - |
| IPNN | 0.8001 | 0.4662 | **0.7850** | 0.3857 | 2 | - |
| OPNN | 0.6630 | 0.5210 | 0.7840 | 0.3852 | 2 | - |
| PNN* | 0.7991 | 0.4699 | 0.7845 | 0.3869 | 2 | - |
| DeepFM | 0.7928 | 0.4584 | 0.7840 | 0.3777 | 2 | - |
| AutoInt | 0.8021 | 0.4511 | 0.7836 | 0.3778 | 2 | 2 |
| DeepEnFM | **0.8077** | **0.4444** | 0.7847 | **0.3700** | 2 | 2 |

Table 2: The ablation study on Criteo dataset. In the first part, we study the contribution of each module in enocder. In the second part, we evaluate the major component. In the third part, we evaluate the variant architectures.

| Model | AUC ($\uparrow$) | Logloss ($\downarrow$) |
|---|---|---|
| DeepEnFM w/o Maxpool | 0.8069 | 0.4448 |
| DeepEnFM w/o Bilinear | 0.8025 | 0.4496 |
| DeepEnFM w/o MHSA | 0.8016 | 0.4511 |
| DeepEnFM w/o FF | 0.8057 | 0.4459 |
| DeepEnFM w/o LN | 0.8075 | 0.4453 |
| DeepEnFM w/o Maxpool & Bilinear | 0.8029 | 0.4496 |
| Encoder + Deep | 0.8059 | 0.4456 |
| Encoder + FM | 0.8037 | 0.4485 |
| Encoder for Deep | 0.7965 | 0.4622 |
| Encoder for Both | 0.8064 | 0.4461 |
| Encoder in Parallel | 0.8075 | **0.4440** |
| DeepEnFM (Full model) | **0.8077** | 0.4444 |

## 4.1 RESULTS

We evaluate the CTR with the standard settings on Criteo and Avazu dataset, as shown in Tab. 1. We highlight the following observations. (1) Our model achieves the best performance of both AUC and LogLoss metrics on Criteo. Specifically, on Criteo our DeepEnFM outperforms the AutoInt by 0.56%, DeepFM by 1.49% on AUC respectively. And DeepEnFM also reduces 0.0067 and 0.0140 on the Logloss comparing to AutoInt and DeepFM respectively. These strongly demonstrate the efficacy of our bilinear encoder. On Avazu dataset, our model achieves the second place in AUC and the first place in Logloss. While, DeepEnFM has still outperformed AutoInt and DeepFM on AUC and Logloss, which indicates the efficacy of our bilinear attention and encoder. (2) The improvement of DeepFM over FM shows the contribution of implicit high-order information in DNN part. (3) The gap between FM and LR reflects that the explicit second order interaction information is important for the CTR task. In summary, all of the implicit high-order information, explicit second order interaction information, and attentional information are critical for the CTR task.

## 4.2 ABLATION STUDY

We conduct extensive ablation study of DeepEnFM as follows: (1) the contribution of each module in the encoder of DeepEnFM; (2) the impact of encoder position; (3) the importance of the depth of encoder layers; (4) the influence of number of heads in multi-head self-attention module.

**Modules in DeepEnFM encoder.** To study the contribution of each modules, we propose several variants of our DeepEnFM. (1) 'DeepEnFM w/o Maxpool': the outputs in MHSA are merged by concatenation instead of max-pooling. (2) 'DeepEnFM w/o Bilinear': the similarity score is calculated by scaled dot product instead of bilinear function. (3) 'DeepEnFM w/o. MHSA': the encoder is implemented without multi-head self-attention module. (4) 'DeepEnFM w/o PF': the encoder is

implemented without point-wise feed-forward module. (5) 'DeepEnFM w/o LN': the encoder is implemented without layer normalization. (6) 'DeepEnFM w/o Maxppol & Bilinear': the encoder is implemented in a standard way.

As shown in the first part of Tab. 2, the performance of 'DeepEnFM w/o. MHSA' is the worst, and 'DeepEnFM w/o. Bilinear' is the second worst, which shows that the our bilinear attention module is critical in DeepEnFM for CTR. Our full model beats all of the variants, which demonstrates the efficacy of each module in encoder.

**Components in DeepEnFM.** To evaluate the effects of each component, we propose two reduced version of our DeepEnFM. (1) 'Enocoder + Deep': the FM component is removed. (2) 'Encoder + FM': the DNN is removed.

The results in the second part of Tab. 2 shows that 'Encoder + Deep' is better than 'Encoder + FM'. The gap to the full model demonstrates the effect of each component.

**Encoder Position.** To evaluate the influence of encoder position, we build several variants with the encoder in different position as shown in Fig. 3. (1) 'Encoder for DNN': we feed the encoder output to DNN, instead to FM; (2) 'Encoder for Both': we feed the encoder output to both the DNN and FM; (3) 'Encoder in Parallel': the original embedding output is fed into DNN, FM and encoder simultaneously. (4) Our DeepEnFM can be viewed as 'Encoder for FM', which uses the encoder to align the embedding for better explicit feature interaction learning.

As shown in the third part of Tab. 2, the results show that our DeepEnFM model 'Encoder for FM' achieves the best results, which indicates that the encoder is more compatible with FM, since both the encoder and FM extract features at vector-wise level. The 'Encoder in Parallel' model achieves the second place, since they can learn in an independent way. The 'Encoder for Both' model is less effective than DeepEnFM, which indicates that serving simultaneously for both the vector-wise and bit-wise learning modules may harm the power of encoder.

Table 3: The performance of models with different number of layers on Criteo dataset.

| Setting/Model | DeepEnFM | | AutoInt | |
|---|---|---|---|---|
| | AUC ($\uparrow$) | Logloss ($\downarrow$) | AUC ($\uparrow$) | Logloss ($\downarrow$) |
| L=0 | 0.7928 | 0.4584 | 0.7934 | 0.4568 |
| L=1 | 0.8049 | 0.4473 | 0.8001 | 0.4510 |
| L=2 | **0.8077** | **0.4444** | 0.8021 | 0.4511 |

**Depth of encoder layers.** We conduct experiments to evaluate the impact of the encoder layers in DeepEnFM, compared with the depth of MHSA layers in AutoInt. Table 3 shows that the performance of DeepEnFM has been improved as the depth of layers grows, and the gain of of the first encoder layer is larger than the second layer. Particularly, we can see the improvement of an encoder layer is larger than a MHSA layer.

Table 4: The performance of models with different number of heads on Criteo dataset.

| Setting/Model | DeepEnFM | | AutoInt | |
|---|---|---|---|---|
| | AUC ($\uparrow$) | Logloss ($\downarrow$) | AUC ($\uparrow$) | Logloss ($\downarrow$) |
| H=1 | **0.8078** | **0.4441** | 0.8010 | 0.4529 |
| H=2 | 0.8077 | 0.4444 | 0.8021 | 0.4511 |
| H=5 | 0.8054 | 0.4466 | 0.8021 | 0.4991 |

**Number of Heads.** We implement DeepEnFM with the head number of 1, 2, 5. Table 4 shows that the model achieves the best results when the number of heads is 1 for DeepEnFM. This is reasonable, the query and key vector dimension reaches the maximum when the number of heads is 1, since the query vector dimension in MHSA is calculated as $\frac{d}{h}$. With more query and key dimension, the bilinear function can gain more generalization ability at the cost of more weights and computation. The result achieves the best balance between the performance and cost when the number of heads is 2 for DeepEnFM. The results of AutoInt show that the influence of head number is smaller than DeepEnFM, because it uses scale dot product function, which is less sensitive to the dimension of query and key vectors.

## 5 CONCLUSIONS

In this paper, we propose a novel framework named Deep neural networks with Encoder enhanced Factorization Machine (DeepEnFM), which aims to learn a better aligned vector embedding through the encoder. The encoder combines the bilinear attention and max-pooling method to gather both the complementary and suppressing information from the content of other fields. The extensive experiments demonstrate that our approach achieves state-of-art performance on Criteo and Avazu dataset.

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
