# OpenReview forum: "DeepEnFM: Deep neural networks with Encoder enhanced Factorization Machine"
_ICLR.cc/2020/Conference — Reject_

### Official Review · AnonReviewer1 · 2019-10-13
**Official Blind Review #1**

**Rating:** 1

**Review:**

The authors propose a model for Click-Through Rate Prediction using a model consisting of an embedding layer, a Transformer stack, a Factorization Machine, and a DNN.

I have several major concerns about the submission:
2. Relevance: This work is extremely application specific, the application is not relevant to this community.
1. Clarity and writing: The contributions which are relevant to the ICLR community are not explained well and the paper needs copy-editing for English grammar
4. Novelty: While seemingly showing good results on some benchmarks, the model is a mix of many components and it's not clear which components actually improve performance and would be worth further study.


Minor comments:

Applying the DNN directly on top of the embeddings, and having a parallel stack of Encoder-FM, is not well explained. What does it mean that "DNN aims at bit-wise level" if the DNN receives the same embedding features as the encoder, which supposedly "learn[s] at vector wise level"?

References to datasets are missing

Ablation study is limited, and has surprising results. E.g. even completely removing self-attention barely makes a dent in how well the method compares to other published work, moving it from rank 1 to rank 2. Otherwise only small tweaks with even more minor effects are made. What about removing e.g. the FM, other major components?

The biggest architectural innovations here are the bi-linear attention mechanism and max-pooling self attention. They are hard to interpret in this context. It's not clear how they would perform in a simpler architecture (e.g. vanilla BERT or Transformer) and in the context of a more standard benchmark. That study would have a lot more relevance to this community than the present one.


**Experience Assessment:**

I do not know much about this area.

**Review Assessment: Checking Correctness Of Derivations And Theory:**

N/A

**Review Assessment: Checking Correctness Of Experiments:**

I assessed the sensibility of the experiments.

**Review Assessment: Thoroughness In Paper Reading:**

N/A

---

> ### Author Response · Authors · 2019-11-15
> **Response to Reviewer 1**
>
> We thank the reviewer for the feedback and address the concerns in detail below
> 1.	Relevance to ICLR
> Thanks, our paper aims to use the encoder to gain better field feature representation for CTR task, which is relevant to learning representations.
> 2.	The meaning of “DNN learns at bit-wise level.”
> The statement “DNN learns at bit-wise level” means that DNN learns the feature representation by the linear and non-linear transformation of neurons. The FM learns the inner product of two features, which is a vector-wise level.
> 3.	Major component study.
> Thanks, we have added the ablation study for the main component. We have removed the FM, DNN to analyze the contribution of FM and DNN in our work. The results show that the AUC of “DeepEnFM w/o DNN” is 0.8037 on criteo dataset, which is higher than AutoInt, which demonstrates the effectiveness of our encoder with bilinear and max-pooling method. Surprisingly, the AUC of  “DeepEnFM w/o FM” is 0.8059 on criteo dataset, which is very close to the full model(0.8077). The gap to the full model demonstrates the effect of each component.

---

### Official Review · AnonReviewer3 · 2019-10-24
**Official Blind Review #3**

**Rating:** 3

**Review:**

This papers proposes DeepEnFM approach for CTR prediction task. In detail, Transformer encoder is applied on top of embeddings to generate new projected embeddings. Such transformer encoder is composed of self-attention with bilinear (to replace dot) and multi-head, which is followed by a mx pooling layer and then a FC layer. Position encoding is utilized then. Besides, some resnet-style trick in placed in the middle. Such encoder output is fed into FM and raw embeddings are feed into DNN part. These two parts are then used for final prediction. Some experimental results show the improvement of the proposed method over other methods.

The major questions are:

*  The assumption of “The field embedding size is very low in CTR” is not reasonable. Do we have any study to verify this hypothesis?
* Regarding to above hypothesis, i think it doesn’t hold for all the CTR prediction tasks. Computation cost will be dramatically increased when embedding size increases because of bilinear between key and query and the FC on top of self-attention.
* The novelty of the proposed method needs to justified to reach the bar of ICLR. The major reason is that 1) the proposed method just replaces MHSA with two changes, i.e., bilinear + max pooling, 2) other tricks such as resnet-style connection, layer norm and position encoding have been adopted everywhere.
* The gain of proposed method is not so clear though the author test to remove each component from the architecture. As the change of encoder part is on top of MHSA, but there is no experiment to show the gain compared to using original MHSA instead of newly proposed bilinear + max pooling. I suggest to do this for better understanding the gain of changes.



**Experience Assessment:**

I have read many papers in this area.

**Review Assessment: Checking Correctness Of Derivations And Theory:**

I assessed the sensibility of the derivations and theory.

**Review Assessment: Checking Correctness Of Experiments:**

I assessed the sensibility of the experiments.

**Review Assessment: Thoroughness In Paper Reading:**

I read the paper at least twice and used my best judgement in assessing the paper.

---

> ### Author Response · Authors · 2019-11-15
> **Response to Reviewer 3**
>
> We thank the reviewer for the feedback and address the concerns in detail below
> 1.	About the novelty.
> The transformer encoder based approaches (e.g. BERT, ViLBERT) have gained great success in NLP and Vision & Language tasks. The application of encoder to CTR is promising to improve the performance by generating context-aware field embedding. And the utilization of encoder with the addition of bilinear and max-pooling method is novel in the CTR task.
>
> 2.	The field embedding size and computation cost
> Thanks, the assumption that “the field embedding size is very low” is made by the observation from works on criteo and Avazu datasets. Actually, the embedding size is controllable, we can project the field feature into a low dimension space, then feed it to the DeepEnFM. With a low embedding size, the computation cost problem of bilinear can be alleviated.
>
> 3.	The experiment of the ordinary MHSA
> The experiment of the ordinary MHSA shows that the AUC is 0.8027 on Criteo data, which also beats the other baselines such as PNN*(0.7991), DeepFM(0.7928), AutoInt(0.8021). This implies that the encoder is effective for CTR task as well as NLP tasks. And the improvement over AutoInt demonstrates the contribution of the point-wise feed-forward layer in the encoder layer. Our DeepEnFM is 0.8077 with the bilinear and max-pooling method.

---

### Official Review · AnonReviewer2 · 2019-10-26
**Official Blind Review #2**

**Rating:** 1

**Review:**

The paper applies Multi-Head Self-Attention (MHSA) to a CTR prediction model with some small changes. The empirical results on two public datasets show it improves performance over some baselines.

First of all, the novelty of the proposed algorithm is limited in that it mainly applies existing mulit-head self-attention. The paper does include some small modifications to MHSA and achieves better performance, such as bi-linear similarity and max-pooling. However, the nature of these changes seems more incremental.

The experiment section is very detailed and the paper conducts several ablation studies to understand which components contribute the most, which is nice. However, the paper is missing several important baselines, for example, Deep & Cross [1], which makes the results less convincing.

Another issue with the paper is that it does not control the model capacity when comparing performance. It is usually the case that increasing model capacity leads to better performance. Given that MHSA and bi-linear similarity have increased a lot of model parameters, it is more fair to compare performance across models with similar capacity. In fact, in [1], they show the logloss on Criteo dataset can be as low as 0.4423 when using large enough parameters.

Minor: in the ablation study, it shows head = 1 has the best performance. In this case, why max-pooling is needed?

Reference:
[1] Wang, R., Fu, B., Fu, G. and Wang, M., 2017, August. Deep & cross network for ad click predictions. In Proceedings of the ADKDD'17 (p. 12). ACM.

**Experience Assessment:**

I have read many papers in this area.

**Review Assessment: Checking Correctness Of Derivations And Theory:**

I assessed the sensibility of the derivations and theory.

**Review Assessment: Checking Correctness Of Experiments:**

I assessed the sensibility of the experiments.

**Review Assessment: Thoroughness In Paper Reading:**

I read the paper at least twice and used my best judgement in assessing the paper.

---

> ### Author Response · Authors · 2019-11-15
> **Response to Reviewer 2**
>
> We thank the reviewer for the feedback and address the concerns in detail below
> 1.	The capacity control problem.
> The capacity problem can be alleviated by projecting the field embedding into a low dimension space and increasing the head number, since the bilinear weight size is $(fieldnum*embsize)^2/headnum$
> 2.	What about Max-pooling when head number is 1.
> The Max-pooling has no effects when the head number is 1. Max-pooling is effective when head_num is bigger than 1. As we mentioned in the ablation study of head_num, though head_num=1 has the best performance, head_num=2 achieves the best balance between the performance and the cost. In the case when head_num=2, max-pooling works.
> 3.	Missing baselines.
> Thanks, Baselines such as xDeepFM, Deep & Cross will be added later.

---

### Decision · Program_Chairs · 2019-12-19

**Decision:**

Reject

**Comment:**

The authors address the problem of CTR prediction by using a Transformer based encoder to capture interactions between features. They suggest simple modifications to the basic Multiple Head Self Attention (MSHA) mechanism and show that they get the best performance on two publicly available datasets.

While the reviewers agreed that this work is of practical importance, they had a few objections which I have summarised below:
1) Lack of novelty: The reviewers felt that the adoption of MSHA for the CTR task was straightforward. The suggested modifications in the form of Bilinear similarity and max-pooling were viewed as incremental contributions.
2) Lack of comparison with existing work: The reviewers suggested some additional baselines (Deep and Cross) which need to be added (the authors have responded that they will do so later).
3) Need to strengthen experiments: The reviewers appreciated the ablation studies done by the authors but requested for more studies to convincingly demonstrate the effect of some components. One reviewer also pointed that the authors should control form model complexity to ensure an apples-to-apples comparison (I agree that many papers in the past have not done this but going froward I have a hunch that many reviewers will start asking for this) .

IMO, the above comments are important and the authors should try to address them in subsequent submissions.

Based on the reviewer comments and lack of any response from the authors, I recommend that the paper in it current form cannot be accepted.